# Sustainable Valorisation of Peach and Apricot Waste Using Green Extraction Technique with Conventional and Deep Eutectic Solvents

**Marina Stramarkou [1,2,*], Vasiliki Oikonomopoulou [1], Margarita Panagiotopoulou [1] , Sofia Papadaki [1] and Magdalini Krokida [1]**

1 Laboratory of Process Analysis and Design, School of Chemical Engineering, National Technical University of Athens (NTUA), 15780 Athens, Greece; vasiaoik@central.ntua.gr (V.O.); sofiachemeng@hotmail.com (S.P.)
2 Institute of Nanoscience and Nanotechnology, National Centre for Scientific Research (NCSR) "Demokritos", 15341 Athens, Greece
* Correspondence: m_stramarkou@hotmail.com

**Abstract:** Worldwide, fruit processing industries reject high volumes of fruit waste, which represent rich sources of phenolic compounds and can be valorised through extraction, and then be reused for food, nutraceutical or cosmetic applications. In the present work, the optimisation of the recovery of phenolic compounds from apricot kernels and pulp, as well as peach pulp, through the green method of ultrasound and microwave assisted extraction (UMAE) is performed. Prior to extraction, a drying step of the pulps is conducted using freeze, vacuum and hot air drying. Except for the conventional extraction solvents of water and ethanol:water, a deep eutectic solvent (DES) formed by choline chloride/urea, and a natural deep eutectic solvent (NaDES) from choline chloride with lactic acid, are used, something that presentsecological benefits. With the aim of discovering the optimum extraction conditions, different values of the parameters of extraction time, utrasonic power and solvent/dry solid ratio are examined, and a mathematical model is developed to correlate them to the extraction yield (EY). The phenolic compounds and the antioxidant activity are determined through UV-Vis spectroscopy and High-Performance Liquid Chromatography (HPLC). The results of the study demonstrated that the most effective solvent in the extraction of apricot kernels is ethanol: water; DES is more efficient in the extraction of apricot pulp and NaDES in the extraction of peach pulps, reaching EYs of 25.65, 26.83 and 17.13%, respectively. In conclusion, both types of fruit waste are proved to have a significant content of valuable compounds, and the use of DES in fruit by-product extraction is effective and seems to be a promising alternative. Thus, the unexploited amounts of waste can be valorised through simple techniques and innovative solvents.

**Keywords:** phenolics; antioxidant activity; ultrasound and microwave assisted extraction; mathematical modelling

## 1. Introduction

Despite the fact that global population growth requires the proper use of agricultural land, it is estimated that one third of agricultural production is wasted. One of the factors that mainly contribute to that environmental burden is the production of fruit and vegetable residues, accounting for 50% of total food waste [1,2]. In fact, the fruit processing industry, an economically predominant industry in Europe, discards 20–50% of the supply volume of the fruit chain as waste [3].

This waste is considered as a degraded substrate and is treated through composting, landfilling or open burning, having a severe effect on environmental pollution [4]. However, the majority of the fruit industry by-products represents a sustainable mine of valuable natural components, which can be utilised and recycled inside the food chain as functional additives in food products [4–6]. Researchers have already examined the use of bioactive

compounds from fruit waste, such as anthocyanins from grapes figs, raspberries and strawberries [7,8], betacyanins from pear [9], caffeic acid and catechin from orange [10] and carotenoids from tomato and pomegranate [11–13].

Among the various fruits, peach and apricot are two widely produced crops worldwide with a high-nutritional value and bio-potential [14]. According to the latest available data of FAOSTAT, Greece is one of the most important producers of peaches and apricots in the world, ranking fifth in the production of peaches with 891 thousand tons in 2020, and eighth in the production of apricots with approximately 126 thousand tons per year [15].

Peach, *Prunus persica* (L.) Batsch, belongs to the Rosaceae family and presents significant antioxidant value thanks to the high content of phenolic compounds, such as flavonoids and anthocyanins, carotenoids and vitamin C [14]. In the same context, apricots, *Prunus armeniaca,* are one of the dietary sources with the highest polyphenol concentration [16]. Phenolic compounds, such as catechin, epicatechin, p-coumaric acid, caffeic acid etc have been identified in apricots [17]. Fruit processing industries use great amounts of peaches and apricots for the production of juices and purees and, finally, dispose high volumes of their types of waste. Nevertheless, the demand for sustainability and the strict legislation lead to the exploitation of these streams as sources of valuable compounds.

The exploitation of fruit residues and the recovery of their nutraceuticals are accomplished through extraction procedures. The used solvent is a process variable with strong effect on the extraction of the target compounds, as well as the quality and safe utilization of the final product [18,19]. When water cannot be used as an extraction medium, the most preferred solvent is ethanol because of its low-boiling point and safe status [3].

However, global research is currently focusing on the substitution of conventional, organic solvents with environmentally benign and cost-effective alternatives. Ionic liquids (ILs) are liquid mixtures between an anion and a cation. They have generated special attention thanks to their unique properties, such as non-volatility, high solubility and ion conductivity, as well as dissolution of hydrophilic or hydrophobic molecules, depending on their properties that can be tuned accordingly. Nonetheless, the use of ionic liquids in food industry as extraction solvents is restricted because of their toxicity and their elevated production costs [3,20,21].

A sub-category of ILs with many similar properties but more distinctive advantages, such as biodegradability and low toxicity, is Deep Eutectic Solvents (DES) and Natural Deep Eutectic Solvents (NaDES) [22]. DES are liquids mixtures of a hydrogen bond acceptor (HBA) and a hydrogen bond donor (HBD), while NaDES are mixtures of a HBA and a HBD from natural sources, such as organic acids, amino acids and sugars, and can be directly used in food formulations [23]. The hard management of DES due to their high viscosity can be answered by tailoring water concentrations or increasing the operating temperature [3,20,21,24]. DES and NaDES constitute a promising alternative of toxic solvents for several chemical and biological applications, especially for DES formed by choline chloride (ChCl) with HBDs, thus lowering the manufacturing cost and elevating their purity. Therefore, in this study, choline chloride with urea and choline chloride with lactic acid diluted with water are examined. Lactic acid is a natural carboxylic acid in milk and can be simply produced by fermentation of carbohydrates [25]. DES and NADES have been used in the extraction of phenolic compounds of mulberries, olive tree leaves, walnut tree leaves, citrus fruits, grapefruits, pears, tomatoes and orange peels, and have shown superior performance compared to conventional solvents [26].

Another determining factor that affects significantly the extraction yields and extracts' bioactivity is the selection of the right extraction technique. Conventional technologies present boundaries, which restrict their implementation, with the most important being the loss of functionality of target compounds due to over-heating and the high consumption of toxic solvents [3,19,27]. Microwave and ultrasound assisted extractions are two of the most employed non-conventional methods for the recovery of phenolic compounds from plant tissues. They are recognised for the low processing times, as well as process acceleration and increasing efficiency [18,28]. Another advantage of microwave and ultrasound assisted

extractions is the ability of many systems to maintain a stable temperature using coolants so that the degradation of thermo-labile molecules is prevented. UAE and MAE have been applied for the recovery of phenolics from many fruit by-products, such as mango, grape, pomegranate, grapefruit, coconut, etc., and has resulted in a 50% higher total phenolic content in comparison with conventional extractions [29].

The improvement of extraction yield using Microwave Assisted Extraction (MAE) is principally attributed to the transformation of the energy of electromagnetic waves into thermal energy, which increases the temperature of the matrix, leading to cellular rupture and, finally, facilitating the diffusion of the compounds in the solvent [30]. In the same framework, the efficiency in Ultrasound Assisted Extraction (UAE) is the result of cavitation forces that lead to the formation of bubbles in the liquid medium. The implosion of the bubbles compresses, adiabatically and rapidly, the gases and vapours within the bubbles or cavities, and afterwards, ruptures the cell walls of the solid matrix. Therefore, the solvent can penetrate easily into the matrix, having enhanced access to cellular content [18].

Prior to extraction, dehydration of biomass with undesirably elevated moisture, not only minimizes the microbial spoilage, but also leads to the tractability of products thanks to their significant volume reduction [31]. Peach and apricot wastes are rich in moisture and therefore drying them during pre-treatment is essential [3]. However, the high concentrations of fruit pulps in heat-labile bioactive compounds require gentle dehydration methods. Hot air drying is cost-effective, and thus, widely used in food industry but with significant disadvantages referring to elevated operating temperatures and nutrient degradation [31,32]. Contrarily, although freeze and vacuum drying demand high-cost equipment and energy consumption, they maintain the sensory and nutritive value of food products, leading to high-quality products [33–35].

The aim of the present work is the optimisation of the recovery of phenolic compounds, from apricot kernels and pulp, as well as peach pulp, through the green method of ultrasound and microwave assisted extraction (UMAE). Prior to extraction, a drying step of the apricot and peach pulp is carried out using freeze drying, vacuum drying or hot-air drying. Except for the conventional extraction solvents of water and ethanol:water, DES formed by choline chloride/urea and NaDES formed by choline chloride with lactic acid, are used in the extractions, thus minimizing the solvent losses and leading to ecological benefits. With the aim of discovering the optimum extraction conditions, the parameters of extraction time, ultrasonic power and solvent/dry-solid ratio are examined. Furthermore, a mathematical modelling is performed in order to correlate the EY with the processing parameters for each raw material and solvent. The qualitative and quantitative determination of the recovered phenolic compounds and the measurement of antioxidant activity is achieved through UV-Vis spectroscopy and high-performance liquid chromatography (HPLC).

## 2. Materials and Methods

### 2.1. Materials and Chemicals

The apricot pulp and kernels, as well as the peach pulp generated by the fruit juice company Aspis Hellenic Juice Industry S.A, were delivered wet at 0 °C in order to prevent a possible degradation. Analytical and HPLC solvents (ethanol, water, acetic acid and methanol) and reagents (choline chloride, urea, lactic acid, Folin–Ciocalteu reagent, radical 2,2-diphenyl-1-picrylhydrazyl, gallic acid and catechin) were purchased from Sigma–Aldrich. The HPLC standards of chlorogenic acid, rutin, quercetin, catechin and epicatechin were purchased from Tech-Line S.A.

### 2.2. Drying Pre-Treatment

Freeze drying and vacuum drying, as well as hot-air drying, were applied individually to apricot and peach pulp in order to reduce the moisture content and compare the efficiency of drying techniques in terms of moisture lowering and bioactive content maintenance. The drying experiments for each drying method were performed separately in triplicate.

Afterwards, the dried apricot and peach pulps, as well as the apricot kernels, were ground to powder so that the extraction efficiency is improved. The exact conditions during drying are detailed below.

### 2.2.1. Freeze Drying

The fruit pulps were immediately placed in special containers with a thickness of 0.5 cm and stored in the freezer (SANYO, MDF-236, Osaka, Japan) at $-30\,^{\circ}$C for 48 h to avoid degradation. Subsequently, freeze drying was carried out using a laboratory freeze dryer (Leybold-Heraeus GT 2A, Koln, Germany) under the effect of 3 mbar vacuum for 5 h.

### 2.2.2. Vacuum Drying

The peach and apricot pulps were placed in special glass containers with a thickness of 0.5 cm in the form of paste. Vacuum drying was performed in the vacuum dryer (Sanyo Gallenkamp PLC, Leicester, England) (220 V, 1000 W at a temperature of 40 $^{\circ}$C and a pressure of $48 \pm 3$ mbar for 5 h).

### 2.2.3. Hot Air Drying

The fruit pulps were dried using a laboratory air dryer (230 V, 1.6 KW). The samples of 0.5 cm thickness were placed on perforated trays perpendicular to the air flow. Hot-air drying was carried out at $50 \pm 2\,^{\circ}$C, 1.0 m/s air velocity and atmospheric pressure for 5 h.

The dried products were stored at $-30\,^{\circ}$C until their further processing. Before extraction, they were milled to achieve a typical particle size of ~0.35 mm.

### 2.2.4. Moisture Content Determination

The calculation of the moisture content (MC) of fresh and dried pulps was accomplished according to the Official Methods of Analysis (OMA- AOAC) [36]. The MC was estimated when the samples reached constant weight using Equation (1):

$$MC\,(\%) = \frac{(W_{init} - W_{final})}{W_{init}} \tag{1}$$

where MC is the moisture content on a wet basis (g $H_2O$/100 g solid), $W_{init}$ is the initial weight before drying (g) and $W_{final}$ is the final weight of samples after drying (g).

The experiments were carried out in triplicates.

### 2.2.5. Drying Kinetics Determination

In order to determine the drying kinetics, the weight of the samples was measured at regular time intervals during drying. A first-order kinetic model was used to describe the moisture loss of the samples with time, since first-order models are more compatible to the experimental data [37,38], as shown in Equation (2):

$$\frac{-dX}{dt} = k\,(X - Xe) \tag{2}$$

where X is the moisture content on a dry basis during drying $((W_{init} - W_{final})/W_{init})$ (g $H_2O$/g dry solid), Xe is the equilibrium moisture of the dried sample (g $H_2O$/g dry solid), k is the drying rate (min$^{-1}$) and t is the drying time (min) [35].

The drying rate was estimated as the slope of the falling-rate drying curve. At zero time, the moisture content on the dry basis of the dry material X (g $H_2O$/g dry solid) was equal to the initial moisture content of the material $X_i$, so the above Equation was integrated and expressed as:

$$Xt = Xe + (Xo - Xe)\exp(-kt) \tag{3}$$

*2.3. Extraction*

The recovery of the phenolic compounds was accomplished through the application of ultrasound and microwave assisted extraction (UMAE). The solvents were selected based on the target compounds desired to be recovered were the following: water, ethanol:water (8:2 *v:v*), the deep eutectic solvent: choline chloride with urea diluted with water (7:3) (DES), and the natural deep eutectic solvent: choline chloride with lactic acid diluted with water (7:3) (NaDES).

2.3.1. Preparation of DES and NADES

Choline chloride, which was used for the preparation of both DES and NADES, was dried in a vacuum dryer (Sanyo Gallenkamp PLC, Leicester, England) (220 V, 1000 W) at 40 °C for 24 h before use. The DES was synthesized using choline chloride and urea at a molar ratio of 1:2, whereas the NADES was synthesized by mixing choline chloride and lactic acid in a molar ratio of 1:2. Both mixtures were prepared in a closed bottle and were heated at 60 °C under magnetic agitation for 2 h until homogeneous, transparent liquids were formed. Finally, they were diluted with water in a volume ratio of 7:3.

2.3.2. Ultrasound and Microwave Assisted Extraction (UMAE)

Fresh and dried apricot and peach pulp samples, as well as apricot kernel samples, were extracted using ultrasounds and microwaves in the XO-SM50 Ultrasonic Microwave Reaction System (Nanjing Xianou Instruments Manufacture Co., Ltd., Nanjing City, China). After preliminary experiments and previous research works [39,40], UMAE experiments were performed under the following operating conditions: ultrasonic frequency 25 kHz, ultrasound emission time/pause time 4.0 s/1.0 s, and microwave power at 250 W. The temperature of the process was controlled during the extraction and remained stable at 40 °C, using a coolant passing through the double-wall extraction beaker so that the risk of degrading thermo-labile components, such as phenolic compounds, is minimised. In addition, the effect of various parameters on the characteristics of the extracts were studied during the extraction. These parameters were the following: the extraction time, the ultrasound power and the solvent/dry-solid ratio. Each independent parameter varied at three levels, which are coded with three numbers: −1, 0, and 1, as presented in Table 1. The specific parameters' ranges were selected in order to compare extreme conditions of extraction time and ultrasound power, as long as they did not degrade the phenolic content of the materials.

**Table 1.** Parameters during Ultrasound- and Microwave-Assisted Extraction (UMAE).

| Parameter | Coding Values/Real Values | | |
|---|---|---|---|
| | **−1** | **0** | **1** |
| Extraction time (min) | 5 | 10 | 20 |
| Ultrasound Power (W) | 150 | 450 | 750 |
| Solvent/dry solid ratio (mL/g) | 10 | 20 | 30 |

2.3.3. Calculation of the Extraction Yield (EY)

The EYs were calculated after the removal of the solvents from the extracts and the determination of the total phenolic content (TPC) using the Folin–Ciocalteu method. Specifically, the aqueous extracts were dried using a freeze dryer (Leybold–Heraeus GT 2A, Koln, Germany), while the organic extracts were placed in a rotary evaporator (Rotavapor R-210, Buchi, Germany). The EYs were expressed as the percentage of the TPC (mg) in the initial dry biomass (g) and were calculated as follows:

$$\text{EY } (\%) = \frac{\text{total phenolic content (mg)}}{\text{initial dry biomass (g)}} \times 100\% \tag{4}$$

### 2.4. Characterisation of Extracts

#### 2.4.1. Total Phenolic Content (TPC)

The total phenolic content of the fresh and dried pulps, as well as the content of the produced extracts, was determined according to Folin–Ciocalteu spectrometric method [41]. Prior to the measurement, 0.5 g of fresh and dried pulps were mixed with 5 mL of ethanol:water (8:2 *v:v*) and left overnight. TPC was expressed as mg of gallic acid equivalent to (GAE)/g of dry mass. The measurements were carried out in duplicate.

#### 2.4.2. Total Flavonoid Content (TFC)

The total flavonoid content was evaluated through the p-Dimethylaminocinnamaldehyde (DMACA) method [42]. Prior to the measurement, 0.5 g of fresh and dried pulps were mixed with 5 mL of ethanol:water (8:2 *v:v*) and left overnight. TFC was expressed as mg of catechin equivalent to (CE)/g dry biomass. The measurements were conducted in duplicate.

#### 2.4.3. Antioxidant Activity (IC50)

The evaluation of the antioxidant activity was carried out using the stable radical 2,2-diphenyl-1-picrylhydrazyl (DPPH) in a solution of 0.003 g DPPH/100 mL of methanol [43]. Prior to the measurement, 0.5 g of fresh and dried pulps were mixed with 5 mL of ethanol:water (8:2 *v:v*) and left overnight. The Radical Scavenging Activity (% RSA), which expresses the ability of the sample to inactivate free radicals, was calculated based on Equation:

$$\%\text{RSA} = \frac{(\text{ABS}_{\text{DPPH}} - \text{ABS}_{\text{mix}})}{\text{ABS}_{\text{DPPH}}}\,100\%\tag{5}$$

where $\text{ABS}_{\text{DPPH}}$ is the absorbance of DPPH solution before the addition of the extract and $\text{ABS}_{\text{mix}}$ is the absorbance of the mixture at 20 min.

The antioxidant activity was expressed through the IC50 (Inhibition Concentration) index, which indicates the concentration of the sample required to inhibit DPPH radical by 50%. Measurements were conducted in duplicate.

#### 2.4.4. Determination of Phenolic Composition by HPLC-DAD

The identification and quantification of phenolic compounds was based on their chromatographic behaviour on High-Performance Liquid Chromatography (HPLC). HPLC analysis was performed with a HPLC Shimandzu HP 1100 Series (USA) equipped with a diode array detector and an automatic Agilent 1200 Series injector. Before injection, the samples were filtered with syringe filters with pore size 0.45 μm and filter size 25 mm. The phenolic compounds were analysed with a Luna C18 column (5 μm × 250 × 4.6 mm) (Phenomenex). The solvents of the mobile phase were 1% vol. acetic acid (solvent A) and methanol (solvent B) and the linear gradient was as follows: 12–25% B from 0 to 15 min, 25–35% B from 15 to 25 min, 35–55% B from 25 to 50 min and 55–65% B from 50 to 60 min, and 65–12% from 60 to 70 min. The operating conditions were as follows: ambient temperature, flow rate of 1 mL/min, detector wavelength of 280 nm and sample injection volume of 20 μL [44]. The identification and quantification of the phenolic compounds, chlorogenic acid, rutin, quercetin, catechin and epicatechin, were carried out by the use of a standard curve formed using solutions of these pure phenolics diluted at known concentrations.

### 2.5. Mathematical Modelling

The prediction of the extraction yield in correlation with the extraction conditions, and specifically the extraction time, the ultrasonic power and the solvent/solid ratio, was accomplished using a simple, mathematical model, which is provided in the following Equation:

$$EY = a_o \left(\frac{t}{t_o}\right)^{a_t} \left(\frac{P}{P_o}\right)^{a_P} \left(\frac{R}{R_O}\right)^{a_R}\tag{6}$$

where $a_o$, $a_t$, $a_P$, $a_R$ are the parameters, $t$ (min) is the extraction time, $P$ (W) is the ultrasonic power, $R$ (mL/g) is the solvent/solid ratio and $t_o$ (min), $P_o$ (W) and $R_o$ (mL/g) are the corresponding values at reference conditions. The reference conditions are the central values (coding values equal to 0 in Table 1), namely, the time equal to 10 min, the ultrasonic power equal to 450 W and the solvent/solid ratio equal to 20 mL/g.

*2.6. Statistical Analysis*

One-way and factorial analyses of variance (ANOVA) were applied to analyse the differences between the obtained extracts concerning their bioactive content. The statistical tests were conducted with STATISTICA software (version 14.0.1., StatSoft®Inc., Palo Alto, CA, USA). Tukey's range test was applied and significantly different values ($p < 0.05$) displayed different superscript [45].

**3. Results**

*3.1. Drying*

Drying of foods not only reduces their weight, but also limits the growth of various microorganisms, thus prolonging their shelf life [37]. The moisture content of apricot pulp and peach pulp was determined according to Equation (1). In addition, the drying kinetics were calculated using the Lewis thin-layer model Equation (3).

The experimental results and the calculated theoretical drying model for all the drying methods of apricot pulp are provided in Figure 1, whereas the drying constants are available in Table 2.

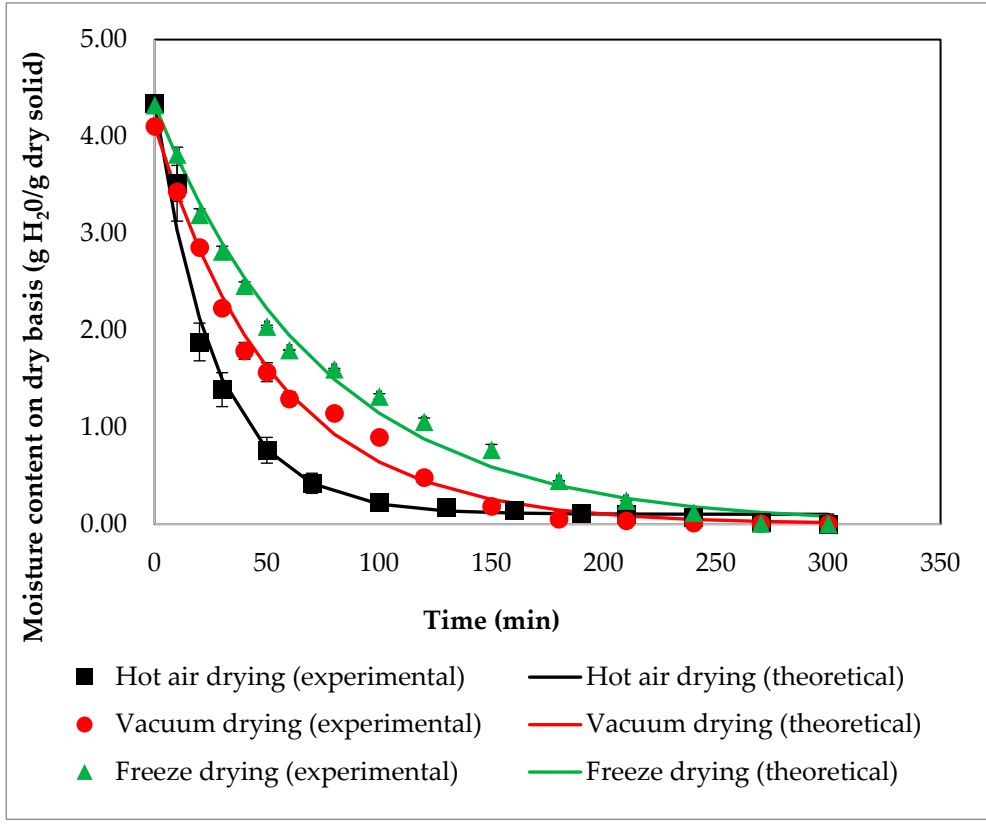

**Figure 1.** Moisture loss of apricot pulp (g $H_2O$/g dry solid) during freeze drying, vacuum drying and hot air drying.

The initial rate of drying is intense due to the small resistance to heat and mass transfer phenomena. However, as the drying process progresses, the rate decreases and dry areas appear on the surface of the material that prevent the heat transfer to the inner layer [46]. The drying time when the equilibrium moisture is reached depends directly on the material

and the drying process, and therefore, different values of drying constants are observed (Table 2).

**Table 2.** Moisture content reduction (expressed in g $H_2O$/100 g solid, %), drying constant (k) (min$^{-1}$), total phenolic content (TPC) (mg gallic acid (GAE)/g dry biomass), total flavonoid content (TFC) (mg catechin (CE)/g dry biomass) and antioxidant activity (IC50) referred to mg of dry biomass for the different types of fresh and dry apricot pulp. Values not sharing the same superscript (separately for each column) are significantly different ($p < 0.05$).

| Biomass Type | Moisture Content Reduction -Wet Basis- (g $H_2O$/100 g Solid) | Drying Constant, $k$ (min$^{-1}$) | TPC (mg GAE/g Dry Biomass) | TFC (mg CE/g Dry Biomass) | IC50 (/mg Dry Biomass) |
|---|---|---|---|---|---|
| Wet | - | - | 23.45 [a] $\pm$ 5.1 $\times$ 10$^{-1}$ | 19.60 [a] $\pm$ 1.4 $\times$ 10$^{-1}$ | 1.60 [a] $\pm$ 3.1 $\times$ 10$^{-2}$ |
| Freeze dried | 80.71 [a] $\pm$ 7.3 $\times$ 10$^{-1}$ | 1.33 $\times$ 10$^{-2}$ [b] $\pm$ 2.3.0 $\times$ 10$^{-3}$ | 30.08 [b] $\pm$ 5.2 $\times$ 10$^{-1}$ | 9.63 [b] $\pm$ 1.3 $\times$ 10$^{-1}$ | 0.98 [b] $\pm$ 3.1 $\times$ 10$^{-2}$ |
| Vacuum dried | 80.55 [a] $\pm$ 5.9 $\times$ 10$^{-1}$ | 1.86 $\times$ 10$^{-2}$ [b] $\pm$ 6.0 $\times$ 10$^{-3}$ | 25.42 [c] $\pm$ 3.4 $\times$ 10$^{-1}$ | 9.29 [b,c] $\pm$ 1.3 $\times$ 10$^{-1}$ | 1.54 [c] $\pm$ 1.3 $\times$ 10$^{-1}$ |
| Hot air dried | 81.29 [a] $\pm$1.1 $\times$ 10$^{-1}$ | 3.71 $\times$ 10$^{-2}$ [a] $\pm$ 8.3.0 $\times$ 10$^{-3}$ | 21.30 [d] $\pm$ 3.9 $\times$ 10$^{-1}$ | 9.18 [c] $\pm$ 1.2 $\times$ 10$^{-1}$ | 1.99 [d] $\pm$ 7.9 $\times$ 10$^{-2}$ |

Except for the moisture content, drying processes cause significant modifications in the physical, chemical and biological characteristics of the samples and may lead to the degradation of the nutritional compounds, especially in the case of high-drying temperatures. Therefore, with the aim of discovering the most appropriate technique for the drying of the apricot pulp, freeze drying, vacuum drying and hot-air drying were compared not only in terms of moisture content reduction, but also in terms of phenolic content, flavonoid content and antioxidant activity. The results are presented in Table 2.

As it is observed, the moisture reduction does not differ significantly between the three drying methods ($p < 0.05$). Freeze drying is the best performing technique considering the phenolic and flavonoid content, and the antioxidant activity. More specifically, the freeze-dried biomass possesses the highest content in bioactive compounds and the lowest IC50 index (best antioxidant capacity). These results are in agreement with previous studies where various drying methods were compared, and it was found that freeze drying represents the mildest method of apricot pulp dehydration, whereas elevated temperatures during drying significantly degrade the bioactive content [47,48].

Regarding the peach pulp, the moisture loss and the kinetics of freeze drying, vacuum drying and hot-air drying are shown in Figure 2 and Table 3.

**Table 3.** Moisture content reduction (expressed in g $H_2O$/100 g solid, %), drying constant (k) (min$^{-1}$), total phenolic content (TPC) (mg gallic acid (GAE)/g dry biomass), total flavonoid content (TFC) (mg catechin (CE)/g dry biomass) and antioxidant activity (IC50) referred to mg of dry biomass for the different types of fresh and dry peach pulp. Values not sharing the same superscript (separately for each column) are significantly different ($p < 0.05$).

| Biomass Type | Moisture Content Reduction -Wet Basis- (g $H_2O$/100 g Solid) | Drying Constant, $k$ (min$^{-1}$) | TPC (mg GAE/g Dry Biomass) | TFC (mg CE/g Dry Biomass) | IC50 (/mg Dry Biomass) |
|---|---|---|---|---|---|
| Wet | - | - | 11.40 [a] $\pm$ 3.1 $\times$ 10$^{-1}$ | 5.02 [a] $\pm$ 1.4 $\times$ 10$^{-1}$ | 1.55 [a] $\pm$ 5.0 $\times$ 10$^{-2}$ |
| Freeze dried | 89.54 [a] $\pm$ 8.1 $\times$ 10$^{-1}$ | 1.61 $\times$ 10$^{-2}$ [b] $\pm$ 4.0 $\times$ 10$^{-3}$ | 19.31 [b] $\pm$ 2.1 $\times$ 10$^{-1}$ | 8.12 [b] $\pm$ 1.3 $\times$ 10$^{-1}$ | 1.33 [a] $\pm$ 5.1 $\times$ 10$^{-2}$ |
| Vacuum dried | 89.02 [a] $\pm$ 4.3 $\times$ 10$^{-1}$ | 1.93 $\times$ 10$^{-2}$ [b] $\pm$ 4.0 $\times$ 10$^{-3}$ | 16.64 [c] $\pm$ 2.3 $\times$ 10$^{-1}$ | 5.59 [b] $\pm$ 1.3 $\times$ 10$^{-1}$ | 2.55 [b] $\pm$ 2.7 $\times$ 10$^{-2}$ |
| Hot air dried | 90.25 [a] $\pm$ 1.9 $\times$ 10$^{-1}$ | 2.81 $\times$ 10$^{-2}$ [a] $\pm$ 6.1 $\times$ 10$^{-3}$ | 13.19 [d] $\pm$ 3.1 $\times$ 10$^{-1}$ | 4.04 [d] $\pm$ 1.2 $\times$ 10$^{-1}$ | 3.56 [c] $\pm$ 5.0 $\times$ 10$^{-2}$ |

Freeze drying prevails over the other drying methods since it exhibits the highest bioactive content and antioxidant capacity. This satisfactory performance of freeze drying is due to several factors. The first factor is the condition of low-drying temperature and oxygen deficiency, which prevent the loss of phenolic compounds because of degradation and isomerization [49]. The second factor concerns the formation of porous structures upon the removal of the ice crystals, which favours the penetration of the solvent inside the biomass [50]. The results of the study are in agreement with the work of Plazzotta (2020), where fresh and dried peach by-products were extracted, and the extracts of the dried samples exhibited double values of phenolic compounds compared with the extracts

of fresh samples, due to the creation of a porous structure and the increase of the contact surface [51].

Thanks to the abovementioned reasons, freeze-dried apricot and peach pulps, along with apricot kernels, were selected to be extracted through UMAE using the solvents of water, ethanol:water (8:2 *v:v*), choline chloride/urea diluted with water (7:3) (DES) and choline chloride/lactic acid diluted with water (7:3) (NaDES).

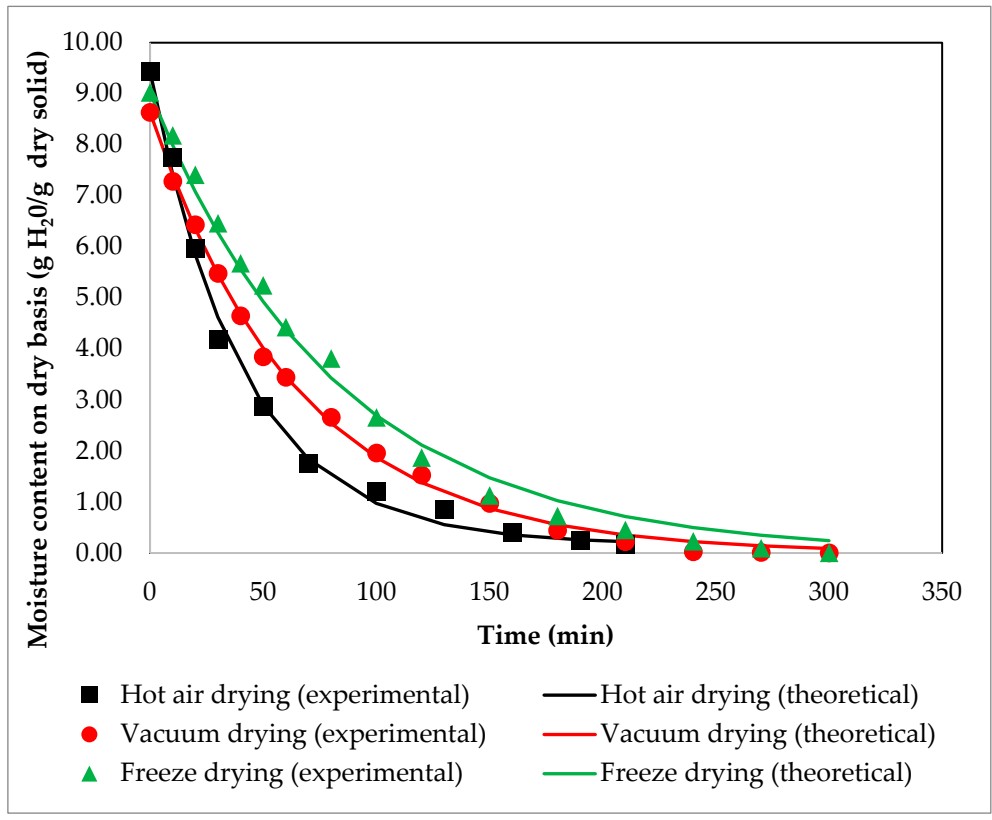

**Figure 2.** Moisture loss of peach pulp (g $H_2O$/g dry solid) during freeze drying, vacuum drying and hot-air drying.

## *3.2. Extraction*

### 3.2.1. Extraction Yield (EY)

The determination of the EY is valuable, since the aim of the study is the optimization of the extraction process and the discovery of the most efficient solvents and extraction conditions. The extraction efficiency and the antioxidant capacity of the extracts are directly dependent on the nature of the solvent, the moisture content of the material to be extracted and the extraction method. Fruits present different compounds with varying chemical characteristics and properties, so a suitable solvent is required for their recovery. The solvents were selected based on the target compounds desired to be recovered and were: water, ethanol:water (8:2 *v:v*), the deep eutectic solvent: choline chloride/urea diluted with water (7:3) (DES) and the natural deep eutectic solvent: choline chloride/lactic acid diluted with water (7:3) (NaDES). The trends of the EYs of the apricot kernel, as a function of the extraction solvents and the extraction parameters, are as follows: ultrasound power, extraction time and solvent/dry-solid ratio, as shown in Figure 3. Table 4 demonstrates the estimated values of the mathematical model that was fitted to the experimental data.

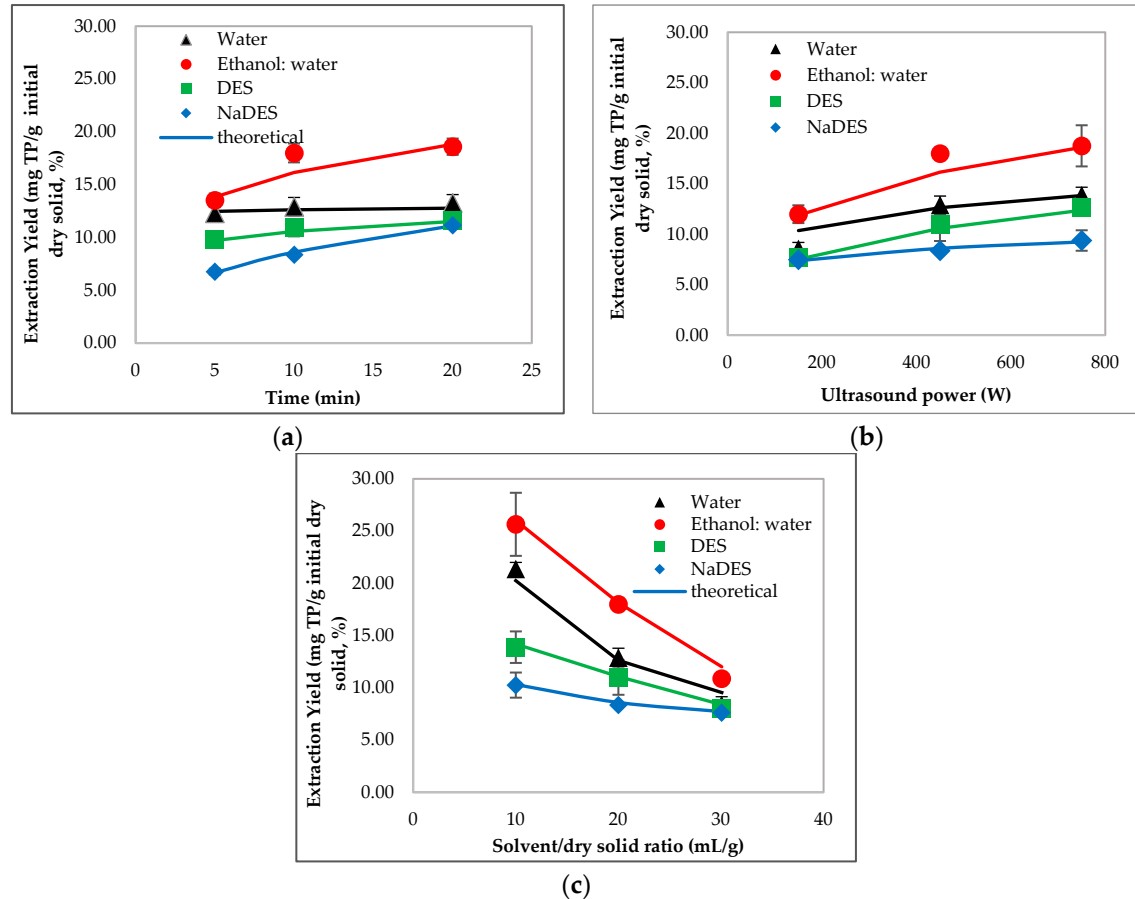

**Figure 3.** Correlation of the extraction yield, EY, (expressed in mg phenolic content/g dry initial solid, %) of apricot kernels with the solvents of water, ethanol:water (8:2 *v:v*), deep eutectic solvent (choline chloride/urea):water (7:3) (DES) and natural deep eutectic solvent (choline chloride/lactic acid):water (7:3) (NaDES) with the factors of: (**a**) time (min), (**b**) ultrasonic power (W) and (**c**) solvent/dry-solid ratio (mL/g).

**Table 4.** Estimated parameters of the extraction yields (EYs) of the apricot kernel: extraction time parameter ($a_t$), ultrasonic power parameter ($a_P$) and solvent/dry-solid ratio parameter ($a_R$).

|  | *a* | $a_t$ | $a_P$ | $a_R$ | $R^2$ (%) |
|---|---|---|---|---|---|
| **Water** | $12.34 \pm 1.98 \times 10^{-1}$ | $5.99 \times 10^{-2} \pm 4.04 \times 10^{-2}$ | $2.96 \times 10^{-1} \pm 4.00 \times 10^{-2}$ | $-8.01 \times 10^{-1} \pm 3.75 \times 10^{-2}$ | 95.03 |
| *p*-value | 0.00 | 0.25 | 0.00 | 0.00 | |
| **Ethanol: water** | $16.14 \pm 4.59 \times 10^{-1}$ | $2.22 \times 10^{-1} \pm 7.11 \times 10^{-2}$ | $2.79 \times 10^{-1} \pm 7.03 \times 10^{-2}$ | $-6.88 \times 10^{-1} \pm 6.91 \times 10^{-2}$ | 88.97 |
| *p*-value | 0.00 | 0.00 | 0.00 | 0.00 | |
| **DES** | $10.58 \pm 2.51 \times 10^{-1}$ | $1.23 \times 10^{-1} \pm 6.11 \times 10^{-2}$ | $3.10 \times 10^{-1} \pm 6.11 \times 10^{-2}$ | $-4.25 \times 10^{-1} \pm 6.48 \times 10^{-2}$ | 93.96 |
| *p*-value | 0.00 | 0.25 | 0.00 | 0.00 | |
| **NaDES** | $8.60 \pm 1.51 \times 10^{-1}$ | $3.68 \times 10^{-1} \pm 4.38 \times 10^{-2}$ | $1.38 \times 10^{-1} \pm 4.05 \times 10^{-2}$ | $-2.62 \times 10^{-1} \pm 5.05 \times 10^{-2}$ | 95.63 |
| *p*-value | 0.00 | 0.00 | 0.00 | 0.00 | |

The results of the analysis show that the lowest recovery of phenolic compounds occurs when the minimum extraction time (5 min), power (150 W) and the maximum solid–solvent ratio (30 mL/g) are applied. On the other hand, better EYs are obtained when the most extreme conditions of time (20 min), power (750 W) and lower ratios are used. The calculation of the EYs and the mathematical modelling demonstrate that among the studied parameters, the most critical ones are the solvent–solid ratio regarding the extractions with water, ethanol:water and DES, and the extraction time regarding the NaDES. This means that the easiest way to improve the extraction efficiency is through the reduction of the ratio by the addition of less solvent to extract the same amount of initial biomass. This

leads to the use of lower amounts of organic and deep eutectic solvents for the valorisation of fruit by-products, which presents both environmental and economic benefits. The most optimal solvent–solid ratio is 10 mL/g ad; when the ratio is higher than this value, the solvent, which is in abundance, absorbs phenolic compounds from the dissolved phase and not from the mass of the apricot kernels.

Furthermore, the extension of the extraction time can slightly improve the EY since the longer the material is in contact with the solvent the more they interact. The positive effect of time on the content of phenolic compounds is also observed in other studies and is attributed to the enhancement of the permeability of the matrix tissues, due to ultrasound cavitation with the increase of processing duration, which leads to the further release of components into the solvent [39,52]. However, preliminary experiments with longer extraction times showed that the yield was increased only in the first 20 min, while thereafter, the extraction rate decreased and the EY reached a plateau. A typical example is in the case of ethanol:water extraction, where the EY after 30 min was 18.91%, after 40 min was 19.03% and after a one-hour extraction was 18.48%. This phenomenon is attributed to the fact that the extraction takes place in two stages. The first stage is characterized by a fast rate and involves the penetration of the solvent into the cell structure and the dissolution of the components in the solvent. The second stage involves the external diffusion of the soluble components through the porous structure of the material and their transfer from the solution that is in contact with the components to the main solution volume. The high EY in short processing times is a result of the acceleration of the wetting phenomenon of the material, due to the collapse of the cell wall, which in turn results in the intensification of mass transfer [53]. Additionally, the extraction for extended times should be avoided in cases that the temperature cannot be controlled and rises significantly; this is something that may lead to the degradation of the phenolic compounds. Therefore, the results of the preliminary experiments with higher extraction times are not included in the study and in the mathematical modelling.

Regarding the ultrasound power, it is discovered that by increasing the values of this parameter, the recovery of phenolic compounds is improved with the power of 750 W showing the best performance, as presented in Figure 3b. However, the fact that the ultrasound power does not affect, to a large extent, the extraction yield, it becomes positive since there is an overall saving of energy and resources, and gentler conditions are preferred as long as they achieve satisfactory results.

Comparing the extraction solvents, the EYs of the aqueous extracts are lower than the extracts of ethanol:water, proving that the solvent system favours the extraction. DES and NADES extracts present even lower EYs. Microwaves increase the temperature during the extraction, thus reducing the viscosity of the solvent and facilitating its penetration into the material and the recovery of the target substances [54]. However, despite the microwaves' temperature and the dilution with water, the viscosity of DES and NADES is still greater than that of the ethanol:water system, thus hindering their efficiency.

Finally, it is worth to mention that the mathematical modelling incomparably adapts to the experimental extraction data, as confirmed by the satisfactory values of the coefficient $R^2$. The respective trends of the EYs of the apricot pulp are shown in Figure 4, whereas Table 5 shows the estimated values of the mathematical model.

The above Figures reveal that the dilution of DES and NaDES with water, and the use of solvent systems in the extractions, achieve the highest recovery of phenolic compounds. The dilution of DES and NaDES with water contributes to the fine-tuning of DES properties with the reduction of their viscosity, which is essential for using viscous DES, leading to improvements in the extraction efficiency [55]. It is interesting to mention that while the DES showed intermediate EYs in the extraction of apricot kernels, this solvent has superior performance in the case of apricot pulp and this may be because of pulp drying. Drying causes rupture and destruction of the cell walls, creating large cavities and intercellular spaces into which the solvent can enter, entrain and recover the target components, thus favoring the extraction [56]. This result is in accordance with the study of Vorobyova

et al. (2021), who report that the extract with the DES of choline chloride with lactic acid has approximately double the content of phenolic compounds than the extract with ethanol:water [57].

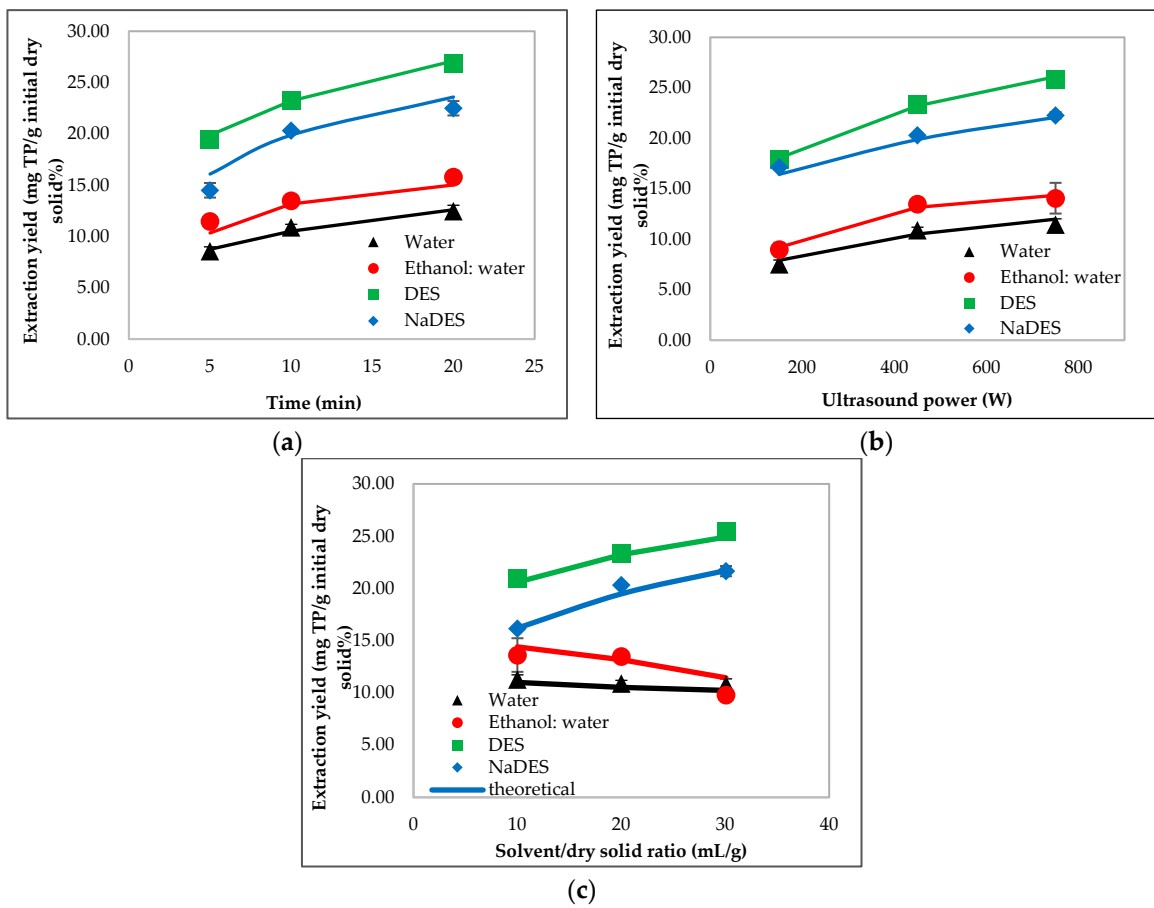

(a)   (b)

(c)

**Figure 4.** Correlation of the extraction yield, EY, (expressed in mg phenolic content/g dry initial solid, %) of apricot pulp with the solvents of water, ethanol:water (8:2 *v:v*), deep eutectic solvent (choline chloride/urea):water (7:3) (DES) and natural deep eutectic solvent (choline chloride/lactic acid):water (7:3) (NaDES) with the factors of: (**a**) time (min), (**b**) ultrasonic power (W) and (**c**) solvent/dry-solid ratio (mL/g).

**Table 5.** Estimated parameters of the extraction yields (EYs) of the apricot pulp: extraction time parameter ($a_t$), ultrasonic power parameter ($a_P$) and solvent/dry-solid ratio parameter ($a_R$).

| | a | $a_t$ | $a_P$ | $a_R$ | $R^2$ (%) |
|---|---|---|---|---|---|
| **Water** | $10.51 \pm 1.28 \times 10^{-1}$ | $2.63 \times 10^{-1} \pm 3.14 \times 10^{-2}$ | $2.59 \times 10^{-1} \pm 3.08 \times 10^{-2}$ | $-6.49 \times 10^{-2} \pm 3.83 \times 10^{-2}$ | 94.12 |
| *p*-value | 0.00 | 0.00 | 0.00 | 0.12 | |
| **Ethanol: water** | $12.47 \pm 3.80 \times 10^{-1}$ | $2.69 \times 10^{-1} \pm 7.80 \times 10^{-2}$ | $2.77 \times 10^{-1} \pm 7.74 \times 10^{-2}$ | $-2.08 \times 10^{-1} \pm 9.06 \times 10^{-2}$ | 91.83 |
| *p*-value | 0.00 | 0.00 | 0.00 | 0.00 | |
| **DES** | $23.21 \pm 1.16 \times 10^{-1}$ | $2.24 \times 10^{-1} \pm 1.31 \times 10^{-2}$ | $2.29 \times 10^{-1} \pm 1.26 \times 10^{-2}$ | $1.73 \times 10^{-1} \pm 1.72 \times 10^{-2}$ | 98.67 |
| *p*-value | 0.00 | 0.00 | 0.00 | 0.00 | |
| **NaDES** | $19.48 \pm 3.26 \times 10^{-1}$ | $2.76 \times 10^{-1} \pm 4.33 \times 10^{-2}$ | $1.55 \times 10^{-1} \pm 4.00 \times 10^{-2}$ | $2.66 \times 10^{-1} \pm 5.87 \times 10^{-2}$ | 88.96 |
| *p*-value | 0.00 | 0.00 | 0.00 | 0.00 | |

Another significant observation, is that while the performance of water and ethanol:water extractions improves by decreasing solvent/solid ratio, DES and NaDES show the opposite behaviour, probably due to the aggregation of the dry-pulp powder and the reduced contact surface of the sample with the solvent, which becomes more noticeable in the case of solvents with higher viscosity.

The respective results of the EYs trends and the mathematical modelling for peach pulp are shown in Figure 5 and Table 6.

**Table 6.** Estimated parameters of the extraction yields (EYs) of the peach pulp: extraction time parameter ($\alpha_t$), ultrasonic power parameter ($a_P$) and solvent/dry-solid ratio parameter ($a_R$).

| | $a$ | $a_t$ | $a_P$ | $a_R$ | $R^2$ (%) |
|---|---|---|---|---|---|
| **Water** | $8.13 \pm 3.12 \times 10^{-2}$ | $2.56 \times 10^{-1} \pm 9.89 \times 10^{-2}$ | $1.20 \times 10^{-1} \pm 8.92 \times 10^{-2}$ | $-5.92 \times 10^{-1} \pm 1.21 \times 10^{-2}$ | 96.89 |
| *p*-value | 0.00 | 0.00 | 0.00 | 0.00 | |
| **Ethanol: water** | $15.05 \pm 2.56 \times 10^{-1}$ | $2.14 \times 10^{-1} \pm 4.38 \times 10^{-2}$ | $1.14 \times 10^{-1} \pm 3.92 \times 10^{-2}$ | $-2.63 \times 10^{-1} \pm 4.96 \times 10^{-2}$ | 92.65 |
| *p*-value | 0.00 | 0.05 | 0.00 | 0.00 | |
| **DES** | $15.35 \pm 1.99 \times 10^{-1}$ | $1.17 \times 10^{-1} \pm 3.41 \times 10^{-2}$ | $2.34 \times 10^{-} \pm 3.26 \times 10^{-2}$ | $-4.56 \times 10^{-1} \pm 4.14 \times 10^{-2}$ | 93.84 |
| *p*-value | 0.00 | 0.05 | 0.00 | 0.30 | |
| **NaDES** | $13.90 \pm 1.02 \times 10^{-1}$ | $3.12 \times 10^{-1} \pm 1.87 \times 10^{-2}$ | $1.11 \times 10^{-1} \pm 1.68 \times 10^{-2}$ | $-2.03 \times 10^{-1} \pm 2.18 \times 10^{-2}$ | 97.62 |
| *p*-value | 0.00 | 0.00 | 0.00 | 0.00 | |

As in the case of the apricot pulp, the comparison of the four solvents used during the extraction of peach pulp demonstrates that water achieves lower EYs. On the other hand, the EYs of the remaining three solvent systems have similar values, with the ethanol:water system presenting the best picture.

However, in order to select the best performing conditions, it is also important to determine the antioxidant capacity of the extracts and to investigate if the values of the antioxidant capacity are in agreement with the profile of the phenolic compound content.

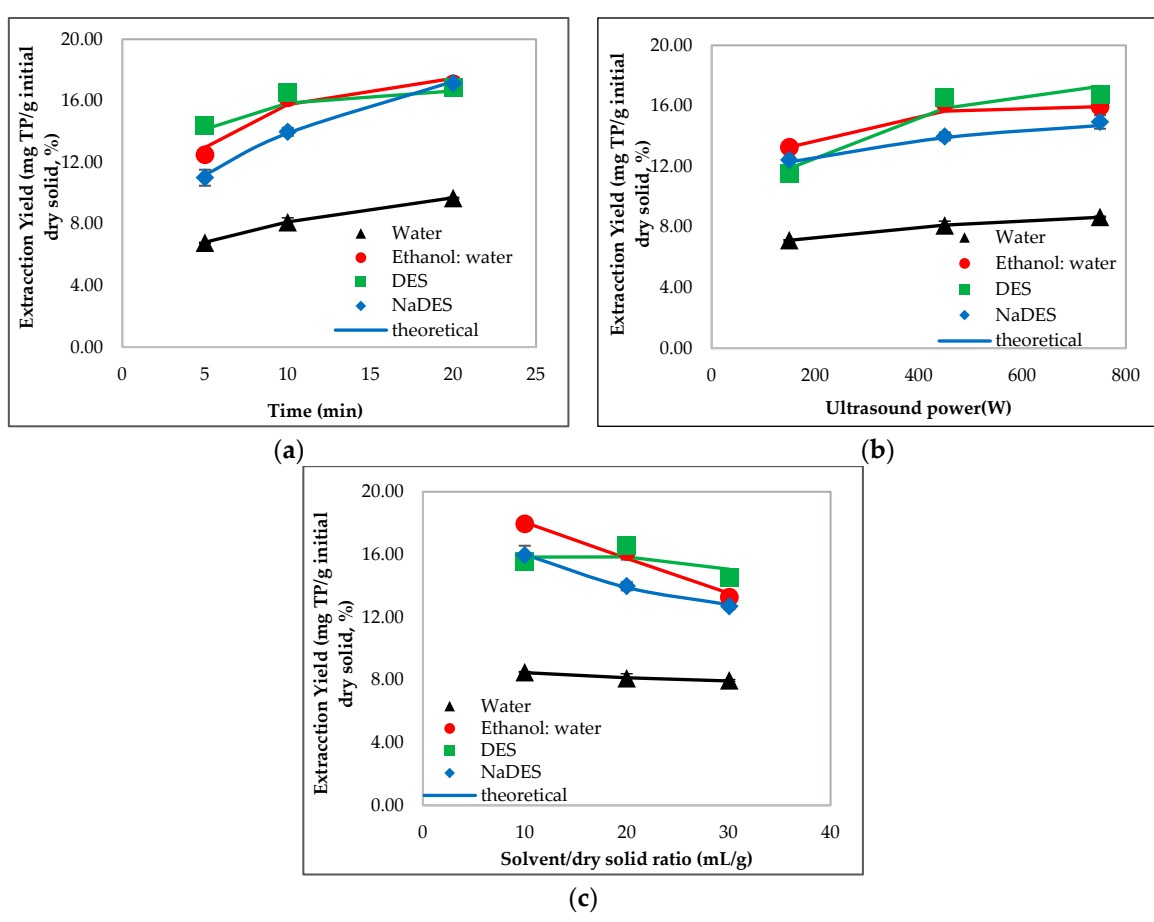

**Figure 5.** Correlation of the extraction yield, EY, (expressed in mg phenolic content/g dry initial solid, %) of peach pulp with the solvents of water, ethanol:water (8:2 *v:v*), deep eutectic solvent (choline chloride/urea):water (7:3) (DES) and natural deep eutectic solvent (choline chloride/lactic acid):water (7:3) (NaDES) with the factors of: (**a**) extraction time (min), (**b**) ultrasonic power (W) and (**c**) solvent/dry-solid ratio (mL/g).

3.2.2. Antioxidant activity

The results of the total phenolic content (mg GAE/g of initial dry solid) and the antioxidant capacity (in mg of initial dry solid) for the central values of the experimental design of the extracts of apricot kernels, apricot pulp and peach pulp, are shown in Tables 7–9, respectively.

As it is observed from the above Tables, the correlation of the total phenolic load and the antioxidant activity is satisfactory, and this is confirmed by the correlation coefficients, whose values are close to −1 with the exception of the apricot kernel extraction with DES. This means that the higher the phenolic load of the extracts, the lower the IC50 index, and therefore, the higher the antioxidant capacity. Among the three raw materials, apricot kernels show the best antioxidant activity, while among the used solvents, the ethanol:water and NaDES extracts have the highest antioxidant activity.

For a better analysis of the extracts' bioactive content, the most important phenolic compounds for the most effective extract of each solvent and each raw material are identified through HPLC-DAD.

3.2.3. Determination of Phenolic Composition by HPLC-DAD

The most abundant and significant phenolic compounds found in apricots and peaches are: chlorogenic acid, catechin, epicatechin, rutin and quercetin [57–59]. The contents of these phenolic compounds for the best performing extracts of apricot kernels, apricot pulp and peach pulp, are displayed in Tables 10–12, respectively.

In general, fruits present complex mixtures of polyphenols. Phenolic substances in fruits are mainly phenolic acids and flavonoids. Apricot is considered a good source of these compounds and has been studied worldwide because its concentration in several phenolic compounds, such as flavanols, chlorogenic acid and rutin, is stable at all ripening stages of all apricot varieties [59].

In the present research, the two main and studied compounds in apricot are rutin and chlorogenic acid, while chlorogenic acid and catechin are the principal compounds in peaches. These findings are in agreement with the results of other studies, where the analyses of the phytochemicals in apricots and peaches showed significant amounts of the above mentioned phenolic compounds [59–62]. Rutin is an antioxidant with many interesting pharmaceutical effects. There are indications that it protects plants from ultraviolet radiation, and that due to its antioxidant action, it has positive effects on human health by reducing blood pressure and protecting the kidneys. On the other hand, chlorogenic acid is an important antioxidant that can contribute to the treatment of atherosclerosis and can reduce cholesterol levels [17].

Concerning the other studied phenolic compounds, their content is different in each raw material extracted with a different solvent. The most effective solvent is ethanol:water in the case of apricot kernels, while DES is more efficient in the extraction of apricot and peach pulps. These results are in accordance with the total phenolic content and the antioxidant activity presented in the previous section. However, it is significant to mention that the separation of DES from the extract is difficult due to its low vapor pressure, which prevents its evaporation and requires a subsequent liquid–liquid extraction using another solvent, or solid–liquid extraction using either resin or molecular sieves or precipitation with the addition of anti-solvents (preferably water) [63].

**Table 7.** Total phenolic content (expressed in mg GAE/g initial dry solid) and antioxidant activity (IC50, referring to mg initial dry solid) of the extracts of apricot kernels. Values not sharing the same superscript (separately for each column) are significantly different ($p < 0.05$).

| Extraction Conditions | | | Water | | Ethanol: Water | | DES | | NaDES | |
|---|---|---|---|---|---|---|---|---|---|---|
| Solvent/Dry Solid Ratio (mL/g) | Time (min) | Ultrasound Power (W) | TPC (mg GAE/g Init. Dry Sol.) | IC50 (/mg Init. Dry Sol.) | TPC (mg GAE/g Init. Dry Sol.) | IC50 (/mg Init. Dry sol.) | TPC (mg GAE/g Init. Dry Sol.) | IC50 (/mg Init. Dry Sol.) | TPC (mg GAE/g Init. Dry Sol.) | IC50 (/mg Init. Dry Sol.) |
| 10 | 10 | 450 | 21.37 [a] ± 0.01 | 0.52 [a] ± 0.06 | 25.65 [a] ± 3.02 | 0.11[a] ± 0.05 | 13.89 [a] ± 1.51 | 2.02 [a] ± 0.25 | 10.26 [a,d] ± 1.21 | 2.25 [a] ± 0.05 |
| 20 | 5 | 450 | 12.29 [b] ± 0.51 | 2.10 [b,d] ± 0.42 | 13.52 [b] ± 0.56 | 0.78 [b] ± 0.06 | 9.83 [b,d] ± 0.41 | 3.12 [b,c] ± 0.21 | 6.76 [b] ± 0.28 | 5.38 [b] ± 0.45 |
| 20 | 10 | 150 | 8.56 [c] ± 0.63 | 3.44 [c] ± 0.38 | 11.98 [b] ± 0.88 | 1.35 [c] ± 0.31 | 7.70 [c] ± 0.56 | 3.70 [b,d] ± 0.25 | 7.47 [b] ± 0.12 | 4.18 [c] ± 0.21 |
| 20 | 10 | 450 | 12.90 [b,d] ± 0.89 | 1.56 [d] ± 0.16 | 18.00 [c] ± 0.58 | 0.37 [a,b] ± 0.09 | 11.00 [b,d] ± 1.67 | 1.70 [a] ± 0.09 | 8.36 [b,c] ± 0.11 | 4.16 [c] ± 0.11 |
| 20 | 10 | 750 | 13.87 [d] ± 0.78 | 2.28 [b] ± 0.05 | 18.76 [c] ± 2.03 | 0.77 [b] ± 0.08 | 12.62 [a,d] ± 0.51 | 2.64 [c] ± 0.12 | 9.38 [a,c] ± 1.02 | 4.02 [c] ± 0.31 |
| 20 | 20 | 450 | 13.28 [b,d] ± 0.21 | 1.58 [d] ± 0.16 | 18.59 [c] ± 0.29 | 0.39 [a,b] ± 0.08 | 11.62 [a,b] ± 0.29 | 2.80 [c] ± 0.12 | 11.16 [d] ± 0.17 | 1.92 [a] ± 0.06 |
| 30 | 10 | 450 | 8.37 [c] ± 0.11 | 3.70 [c] ± 0.21 | 10.88 [b] ± 0.14 | 1.25 [c] ± 0.25 | 8.08[c] ± 0.28 | 3.74 [d] ± 0.36 | 7.62 [b] ± 0.10 | 4.78 [b,c] ± 0.35 |
| TPC- IC50 correlation coefficients | | | −0.88 | | −0.80 | | −0.69 | | −0.85 | |

**Table 8.** Total phenolic content (expressed in mg GAE/g initial dry solid) and antioxidant activity (IC50, referring to mg initial dry solid) of the extracts of apricot pulp. Values not sharing the same superscript (separately for each column) are significantly different ($p < 0.05$).

| Extraction Conditions | | | Water | | Ethanol: Water | | DES | | NaDES | |
|---|---|---|---|---|---|---|---|---|---|---|
| Solvent/Dry Solid Ratio (mL/g) | Time (min) | Ultrasound Power (W) | TPC (mg GAE/g Init. Dry Sol.) | IC50 (/mg Init. Dry Sol.) | TPC (mg GAE/g Init. Dry Sol.) | IC50 (/mg Init. Dry Sol.) | TPC (mg GAE/g Init. Dry Sol.) | IC50 (/mg Init. Dry Sol.) | TPC (mg GAE/g Init. Dry Sol.) | IC50 (/mg Init. Dry Sol.) |
| 10 | 10 | 450 | 11.29 [a] ± 0.28 | 3.76 [a] ± 0.28 | 13.62 [a] ± 1.60 | 3.54 [a] ± 0.08 | 20.96 [a] ± 0.40 | 2.62 [a] ± 0.02 | 16.15 [a] ± 0.07 | 2.10 [a] ± 0.09 |
| 20 | 5 | 450 | 8.58 [b] ± 0.11 | 6.36 [b] ± 0.23 | 11.47 [b,c] ± 0.48 | 4.10 [b] ± 0.06 | 19.50 [b] ± 0.45 | 3.90 [b] ± 0.03 | 14.50 [b] ± 0.71 | 4.66 [b] ± 0.08 |
| 20 | 10 | 150 | 7.53 [b] ± 0.42 | 8.20 [c] ± 0.11 | 8.99 [c] ± 0.66 | 5.10 [c] ± 0.18 | 17.91 [c] ± 0.42 | 4.52 [c] ± 0.06 | 17.15 [a] ± 0.21 | 1.76 [c] ± 0.09 |
| 20 | 10 | 450 | 10.91 [a] ± 0.29 | 3.72 [a] ± 0.21 | 13.50 [a,b] ± 0.43 | 3.32 [a] ± 0.11 | 23.32 [d] ± 0.72 | 0.22 [d] ± 0.02 | 20.32 [c] ± 0.12 | 1.02 [d,e] ± 0.03 |
| 20 | 10 | 750 | 11.46 [a,c] ± 0.57 | 3.96 [a] ± 0.10 | 14.07 [a] ± 1.52 | 2.02 [d] ± 0.13 | 25.90 [e,f] ± 0.29 | 1.78 [e] ± 0.01 | 22.28 [d] ± 0.32 | 0.50 [f] ± 0.10 |
| 20 | 20 | 450 | 12.48 [c] ± 0.71 | 2.46 [d] ± 0.09 | 15.80 [a] ± 0.24 | 0.64 [e] ± 0.02 | 26.83 [e] ± 0.18 | 1.58 [f] ± 0.03 | 22.50 [d] ± 0.71 | 0.90 [d] ± 0.04 |
| 30 | 10 | 450 | 10.77 [a] ± 0.11 | 4.56 [e] ± 0.11 | 9.80 [c] ± 0.13 | 4.02 [b] ± 0.08 | 25.42 [f] ± 0.72 | 0.24 [d] ± 0.06 | 21.65 [d] ± 0.49 | 1.12 [e] ± 0.04 |
| TPC- IC50 correlation coefficients | | | −0.95 | | −0.84 | | −0.81 | | −0.86 | |

**Table 9.** Total phenolic content (expressed in mg GAE/g initial dry solid) and antioxidant activity (IC50, referring to mg initial dry solid) of the extracts of peach pulp. Values not sharing the same superscript (separately for each column) are significantly different ($p < 0.05$).

| Extraction Conditions | | | Water | | Ethanol: Water | | DES | | NaDES | |
|---|---|---|---|---|---|---|---|---|---|---|
| Solvent/Dry Solid Ratio (mL/g) | Time (min) | Ultrasound Power (W) | TPC (mg GAE/g Init. Dry Sol.) | IC50 (/mg Init. Dry Sol.) | TPC (mg GAE/g Init. Dry Sol.) | IC50 (/mg Init. Dry Sol.) | TPC (mg GAE/g Init. Dry Sol.) | IC50 (/mg Init. Dry Sol.) | TPC (mg GAE/g Init. Dry Sol.) | IC50 (/mg Init. Dry Sol.) |
| 10 | 10 | 450 | 8.49 [a] ± 0.03 | 4.60 [a] ± 0.30 | 17.95 [a] ± 0.12 | 0.35 [a] ± 0.09 | 15.55 [a] ± 0.23 | 0.92 [a,e] ± 0.09 | 15.95 [a] ± 0.60 | 1.17 [a] ± 0.05 |
| 20 | 5 | 450 | 6.78 [b] ± 0.11 | 6.84 [b] ± 0.28 | 12.50 [b] ± 0.04 | 3.56 [b] ± 0.35 | 14.39 [b] ± 0.11 | 1.43 [b] ± 0.11 | 11.01 [b] ± 0.52 | 3.12 [b] ± 0.08 |
| 20 | 10 | 150 | 7.13 [b] ± 0.04 | 5.91 [c] ± 0.15 | 13.28 [b] ± 0.29 | 2.02 [c] ± 0.26 | 11.52 [c] ± 0.25 | 5.96 [c] ± 0.15 | 12.43 [c] ± 0.23 | 2.85 [b,c] ± 0.11 |
| 20 | 10 | 450 | 8.11 [c] ± 0.29 | 4.34 [a] ± 0.24 | 16.20 [c] ± 0.52 | 0.51 [a,d] ± 0.06 | 16.56 [d] ± 0.18 | 056 [a] ± 0.09 | 13.98 [d] ± 0.27 | 2.84 [b,c] ± 0.24 |
| 20 | 10 | 750 | 8.66 [a] ± 0.06 | 5.50 [c] ± 0.11 | 15.96 [c] ± 0.28 | 0.93 [d,e] ± 0.06 | 16.77 [d] ± 0.15 | 1.22 [b,e] ± 0.15 | 14.94 [a,d] ± 0.45 | 1.34 [a] ± 0.16 |
| 20 | 20 | 450 | 9.68 [d] ± 0.07 | 2.68 [d] ± 0.13 | 17.10 [d] ± 0.26 | 0.45 [a,d] ± 0.09 | 16.84 [d] ± 0.42 | 1.47 [b] ± 0.21 | 17.13 [e] ± 0.25 | 0.53 [d] ± 0.08 |
| 30 | 10 | 450 | 7.97 [c] ± 0.11 | 5.28 [c] ± 0.30 | 13.28 [b] ± 0.18 | 1.30 [e] ± 0.06 | 14.55 [b] ± 0.51 | 2.04 [d] ± 0.21 | 12.70 [c] ± 0.19 | 2.74 [c] ± 0.11 |
| TPC- IC50 correlation coefficients | | | −0.86 | | −0.84 | | −0.87 | | −0.91 | |

**Table 10.** Content of optimal extracts of apricot kernels in chlorogenic acid, catechin, epicatechin, rutin and quercetin (mg/100 g). Values not sharing the same superscript (separately for each column) are significantly different ($p < 0.05$).

| | Optimal Extraction Conditions | Chlorogenic Acid (mg/100 g) | Catechin (mg/100 g) | Epicatechin (mg/100 g) | Rutin (mg/100 g) | Quercetin (mg/100 g) |
|---|---|---|---|---|---|---|
| Water | 10 (mL/g), 10 min, 450 W | 45.62 [a] ± 0.64 | 6.42 [a] ± 0.11 | 2.09 [a] ± 0.09 | 62.60 [a] ± 0.60 | 4.25 [a] ± 0.21 |
| Ethanol: water | 10 (mL/g), 10 min, 450 W | 46.65 [a] ± 0.82 | 10.36 [b] ± 0.09 | 4.11 [b] ± 0.11 | 71.66 [b] ± 0.71 | 6.05 [b] ± 0.33 |
| DES | 10 (mL/g), 10 min, 450 W | 20.09 [b] ± 0.06 | 6.01 [a] ± 0.41 | 4.01 [b] ± 0.08 | 24.10 [c] ± 0.10 | 1.07 [c] ± 0.06 |
| NaDES | 20 (mL/g), 20 min, 450 W | 18.00 [c] ± 0.30 | 6.41 [a] ± 0.40 | 6.60 [c] ± 0.60 | 22.71 [c] ± 0.51 | 0.82 [c] ± 0.09 |

**Table 11.** Content of optimal extracts of apricot pulp in chlorogenic acid, catechin, epicatechin, rutin and quercetin (mg/100 g). Values not sharing the same superscript (separately for each column) are significantly different ($p < 0.05$).

| | Optimal Extraction Conditions | Chlorogenic Acid (mg/100 g) | Catechin (mg/100 g) | Epicatechin (mg/100 g) | Rutin (mg/100 g) | Quercetin (mg/100 g) |
|---|---|---|---|---|---|---|
| Water | 20 (mL/g), 20 min, 450 W | 14.51 [a] ± 0.55 | 1.38 [a] ± 0.21 | 1.01 [a] ± 0.21 | 15.62 [a] ± 0.14 | 1.23 [a] ± 0.15 |
| Ethanol: water | 20 (mL/g), 20 min, 450 W | 36.31 [b] ± 0.32 | 2.08 [a] ± 0.27 | 2.01 [a,b] ± 0.38 | 55.66 [b] ± 0.32 | 0.99 [a] ± 0.07 |
| DES | 20 (mL/g), 20 min, 450 W | 56.89 [c] ± 0.95 | 9.99 [b] ± 1.02 | 3.69 [c] ± 0.69 | 99.31 [c] ± 2.1 | 5.96 [b] ± 0.15 |
| NaDES | 20 (mL/g), 20 min, 450 W | 41.32 [d] ± 1.37 | 6.66 [c] ± 1.02 | 3.02 [b,c] ± 0.61 | 68.55 [d] ± 0.87 | 4.16 [c] ± 0.54 |

**Table 12.** Content of optimal extracts of peach pulp in chlorogenic acid, catechin, epicatechin, rutin and quercetin (mg/100 g). Values not sharing the same superscript (separately for each column) are significantly different ($p < 0.05$).

| | Optimal Extraction Conditions | Chlorogenic Acid (mg/100 g) | Catechin (mg/100 g) | Epicatechin (mg/100 g) | Rutin (mg/100 g) | Quercetin (mg/100 g) |
|---|---|---|---|---|---|---|
| Water | 20 (mL/g), 20 min, 450 W | 17.32 [a] ± 0.25 | 11.38 [a] ± 0.32 | 1.00 [a] ± 0.10 | 1.89 [a] ± 0.09 | 2.32 [a] ± 0.08 |
| Ethanol: water | 10 (mL/g), 10 min, 450 W | 60.00 [b] ± 1.8 | 46.92 [b] ± 1.47 | 4.01 [b] ± 0.60 | 1.46 [a] ± 0.08 | 7.03 [b] ± 0.15 |
| DES | 20 (mL/g), 20 min, 450 W | 61.00 [b] ± 2.95 | 54.71 [c] ± 2.33 | 4.40 [b] ± 1.29 | 9.32 [b] ± 2.1 | 5.81 [b] ± 1.7 |
| NaDES | 20 (mL/g), 20 min, 450 W | 56.32 [b] ± 2.08 | 41.81 [d] ± 1.33 | 4.02 [b] ± 1.10 | 2.50 [a] ± 0.65 | 3.18 [a] ± 0.65 |

## 4. Conclusions

In conclusion, an extraction technique is considered ideal when it can offer high yields, while being time-efficient, as well as non-destructive to the material. Additionally, its environmental and economic requirements must be as low as possible, so that the food, nutraceutical, cosmetic or packaging industry prefers it as a green option with the potential of achieving a safe and high-quality extract. Thus, in recent years, microwave and ultrasound extractions have been proposed as alternative methods, and deep eutectic solvents as alternative solvents.

During UMAE, the samples are sonicated so that their cell wall is disrupted, and then heated with microwaves so that the penetration of the solvent into the plant tissue is facilitated. In only 20 min of processing, the samples showed substantial EYs, proving

that the combination of methods, such as MAE and UAE, is possible and can significantly reduce the processing time, even with the use of green solvents, such as water. Another noteworthy advantage is the fact that the samples are not destroyed and the remaining biomass can be used in other applications, such as animal feed.

Finally, the by-products of the juice industry can be a rich and sustainable source of natural antioxidants and phenolic compounds. In addition to the pre-treatment process, the extraction, the used solvent and the process conditions during extraction affected the efficiency of the process to a great extent. Extracts recovered with ethanol:water (8:2), and choline chloride/urea diluted with water at a ratio of 7:3 showed high activity against oxidative radicals, while the optimal extraction conditions in the majority of the experiments were atlow solvent–solid ratio, medium ultrasonic power (450 and 750 W) and times from 10 to 20 min. As it is clearly shown in this study, the new proposed extraction methods and solvents can transform a waste or a by-product into a product of increased added value.

**Author Contributions:** Conceptualization, M.K. and M.S.; methodology, M.S and M.P.; software, M.S. and V.O.; validation, M.S., V.O. and M.K.; formal analysis, M.S.; investigation, M.S. and M.P.; resources, M.K and S.P.; data curation, M.S. and V.O.; writing—original draft preparation, M.S.; writing—review and editing, M.S.; visualization, M.S.; supervision, M.K.; project administration, M.K.; funding acquisition, M.S. and M.K. All authors have read and agreed to the published version of the manuscript.

**Funding:** This research is sponsored by Stavros Niarchos Foundation through the Industrial Research Fellowship Program at National Centre for Scientific Research "Demokritos" in collaboration with National Technical University of Athens.

**Acknowledgments:** The authors would like to thank the Hellenic Juice Industry SA for the provision of the fruit waste.

**Conflicts of Interest:** The authors declare no conflict of interest.

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
