# Peer review of "Sustainable Valorisation of Peach and Apricot Waste Using Green Extraction Technique with Conventional and Deep Eutectic Solvents"

_resources, doi:10.3390/resources12060072_

Round 1

Reviewer 1 Report

Dear Authors,

The manuscript reports on the effect of different extraction techniques (ultrasound and microwave-assisted extraction) with water-ethanol, DES and NADES on the extraction yield of antioxidants from peach and apricot waste. Below are a few suggestions for improvement:

1. There are many works reported on TPC/antioxidant extraction available. Please highlight the novelty of the work. 

2. Please discuss the selection of DES and NADES in your study. You may refer to https://doi.org/10.1016/j.jclepro.2022.132239 and https://doi.org/10.1016/j.indcrop.2020.112363

3. The authors are suggested to include some findings obtained from phenolic extraction using conventional solvent, DES, and the incorporation of MAE and UAE in extraction in the Introduction.

4. Please discuss the selection of the parameters during the extraction method and their range. 

5. The methodology can be improved. For example, are the drying processes sequential or only 1 was applied at a time?

6. Referring to Section 2.2.5, please specify why first-order model was applied. Any literature or preliminary work to support the selection? Similarly, any literature to support the selection of the mathematical model?

7. Referring to line 377 - 378, please provide the preliminary results as supporting document. 

8. Referring to lines 421 - 423, do any results/literature support the claim? Please include it in the discussion to avoid speculation. 

9. The authors are strongly encouraged to include a comparison study in order to highlight the strength and weakness of the work. 

Thank you. 

The manuscript is well-organized and written. However, if necessary, the authors may consider sending the manuscript for English proofreading. 

Author Response

Answer to Reviewer 1

We would like to thank the Reviewer for the useful comments made on our manuscript, which helped us to improve it. Please find below the changes made on our manuscript following the Reviewer’s recommendations.

Dear Authors,

The manuscript reports on the effect of different extraction techniques (ultrasound and microwave-assisted extraction) with water-ethanol, DES and NADES on the extraction yield of antioxidants from peach and apricot waste. Below are a few suggestions for improvement:

  1. There are many works reported on TPC/antioxidant extraction available. Please highlight the novelty of the work.

The novelty of this work relies on the combination of novel and green solvents, such as  DESs/ NaDESs, and  environmentally friendly techniques, such as ultrasound and microwave assisted extraction, to extract phenolic compounds from valuable and very common fruit by-products, the volume of which is very high in Europe (since they are largely used in juices), but remain unexploited.  In addition, a complete work is achieved presenting not only the extraction part, but also the effect of various pre-processing techniques in order to optimise the extraction yield.

  1. Please discuss the selection of DES and NADES in your study. You may refer to https://doi.org/10.1016/j.jclepro.2022.132239and https://doi.org/10.1016/j.indcrop.2020.112363

The selection of DES is discussed in Introduction and the suggested references are added in order to better support our choice. Lactic acid is a natural acid and this is why it is selected to be studied.

  1. The authors are suggested to include some findings obtained from phenolic extraction using conventional solvent, DES, and the incorporation of MAE and UAE in extraction in the Introduction.

Sentences describing the use of NADES and DES in various fruits and leaves are added and parts focusing on application of UAE and MAE in fruit by-products compared with conventional techniques are introduced.

  1. Please discuss the selection of the parameters during the extraction method and their range. 

Regarding the operating condition of extraction time, the aim of the study is the application of “green” extractions and the use of environmentally friendly and energy-efficient conditions. Therefore, short extraction durations are chosen between 5 and 20 min to show that even during this short time the UMAE yield can be very high.

Regarding the ultrasonic power, the XO-SM50 Ultrasonic Microwave Reaction System reaches values up to 900 W. During our preliminary experiments, we performed extractions of 30 min and 900 W. These maximum values had a negative impact on phenolic content. Therefore lower power, namely 150, 450 and 750 W (16.7%, 50% and 83% of maximum), and lower duration (5, 10 and 20 min) were selected to be further studied, since they do not deteriorate the quality of the phenolic compounds. The decrease of the TPC when extreme powers and higher durations are applied is mentioned in other scientific papers [https://doi.org/10.1111/jfpp.14636, https://doi.org/10.1016/j.foodchem.2017.08.114,  https://doi.org/10.1016/j.lwt.2017.10.065]. The above-mentioned comments together with related literature to support our range of ultrasonic power utilized is added in the revised manuscript.

Regarding the temperature, which was maintained stable, 40 oC were chosen to prevent the degradation of the thermos-labile molecules.

  1. The methodology can be improved. For example, are the drying processes sequential or only 1 was applied at a time?

The drying techniques were applied separately (one at a time) in order to compare the efficiency of these drying techniques in terms of moisture lowering and bioactive content preservation. An explanatory sentence is added in Materials and Methods.

  1. Referring to Section 2.2.5, please specify why first-order model was applied. Any literature or preliminary work to support the selection? Similarly, any literature to support the selection of the mathematical model?

It is found in the literature that: Dehydration kinetics are typically modeled by fitting the experimental drying curves to (a) empirical thin-layer models (e.g., Wang and Singh, Weibull), (b) semitheoretical ones derived from Newton's (e.g., Lewis, Page, and modified Page) or Fick's second laws (e.g., exponential, two-term, logarithmic, and Henderson and Pabis), and (c) first-order kinetics models (C. Ertekin & Firat, 2017; Krokida & Philippopoulos, 2005; Onwude, Hashim, Janius, Nawi, & Abdan, 2016)  [https://doi.org/10.1111/jfpe.13192]. It is also mentioned in literature that first-order models are more compatible to the experimental data of drying [https://doi.org/10.3390/pr10102082]. Because of the literature data and good results of first- order kinetic models in previous works of the authors [https://doi.org/10.1080/10498850.2021.1900969], first order kinetics, and specifically Peleg model, were chosen.

  1. Referring to line 377 - 378, please provide the preliminary results as supporting document. 

A typical example of the preliminary results is added in the Results and Discussion pointing out the trend of the EY during extraction time. Since the volume of data of the study is very high, the authors decided to provide a very characteristic example of preliminary result to support their claiming.

  1. Referring to lines 421 - 423, do any results/literature support the claim? Please include it in the discussion to avoid speculation. 

In order to avoid speculation, the explanation of lowering the viscosity was given, which is also found in other study [https://doi.org/10.1016/j.molliq.2021.117717]

  1. The authors are strongly encouraged to include a comparison study in order to highlight the strength and weakness of the work. 

Similar works studying the extraction of phenolic compounds are added in the Results and Discussion Section and the findings of their studies (Extraction Yield and HPLC) are compared to our results.

Reviewer 2 Report

I really appreciated the work done on "Sustainable valorization of peach and apricot waste using green extraction technique with conventional and deep eutectic solvents." In my opinion, the work turns out to be well done and with excellent insights for the future application of peach and apricot industrial waste recovery. You will find some minor tips and suggestions for improving the manuscript, which is already in excellent publication condition for me.

Author Response

Answer to Reviewer 2

We would like to thank the Reviewer for spotting some mistakes and for the useful comments made on our manuscript, which helped us to improve it. Please find below the changes made on our manuscript following the Reviewer’s recommendations.

  • The references are added in the introduction
  • “Prunus persica (L.) Batsch” is corrected
  • “Rosaceae family” is corrected
  • “apricots, Prunus armeniaca” is added
  • “waste.”, “.”, “Table 9” are corrected
  • In Line 96 and Line 210 a small sentence was added to describe the ability of our extraction system to maintain stable the temperature by passing coolant through the double wall extraction beaker and by controlling the temperature throughout the extraction by seeing it to the screen of the extraction system that is connected to a thermocouple put in the beaker.
  • All the solvents, reagents and standards of the extraction are mentioned
  • “v:v” is added
  • p” is corrected

Reviewer 3 Report

The article it is well written and has publication potential because presents scientific interest and presents attractive information from a technical point of view. But, the article needs some improvements presented below.

1) Introduction: the authors present the production of peach and apricot in Greece. Authors can specify how much of this total becomes waste;

2) Materials and Methods: a flowchart of the experimental process should be presented to facilitate understanding;

3) Section Drying Kinetics Determination: authors must provide information (references, for example) to justify the choice of first-order kinetics for drying;

4) Section Ultrasound and microwave assisted extraction (UMAE): authors must provide information (references, for example) to justify the choice of the operating conditions;

5) Section Calculation of the Extraction Yield (EY): provide the definition of TPC. In addition, the units in equation 4 (mg and g) must be in the same order of magnitude;

6) Section 3.1. Drying: provide a title for Table 3;

7) Section 3.2.1. Extraction Yield (EY): in ” Table 3 demonstrates the estimated values of the mathematical model that was fitted to the experimental data.” the correct is Table 4. The numbering of the following tables should also be corrected;

8) The results obtained should be compared with similar studies;

9) Conclusion: indicate the best ultrasound power (W) for the operating condition.

Moderate revisions.

Author Response

Answer to Reviewer 3

We would like to thank the Reviewer for the useful comments made on our manuscript, which helped us to improve it. Please find below the changes made on our manuscript following the Reviewer’s recommendations.

The article it is well written and has publication potential because presents scientific interest and presents attractive information from a technical point of view. But, the article needs some improvements presented below.

1) Introduction: the authors present the production of peach and apricot in Greece. Authors can specify how much of this total becomes waste;

Unfortunately, this specific kind of data of peach and apricot waste after the process of juicing, cannot be found even though we looked for it in many official sites like FAOSTAT. However, the information that is available is that in general fruits that are being processed for juices generate by-products, which may account for 20–80% of the whole fruit (10.1016/j.foodchem.2016.12.093, https://doi.org/10.1021/acs.jafc.2c00756).

2) Materials and Methods: a flowchart of the experimental process should be presented to facilitate understanding;

Thank you for your comment. A graphical abstract is presented so that the understanding of the processes is easier. Please inform us if you would like to add another flowchart in Materials and Methods.

3) Section Drying Kinetics Determination: authors must provide information (references, for example) to justify the choice of first-order kinetics for drying;

It is found in the literature that: Dehydration kinetics are typically modeled by fitting the experimental drying curves to (a) empirical thin-layer models (e.g., Wang and Singh, Weibull), (b) semitheoretical ones derived from Newton's (e.g., Lewis, Page, and modified Page) or Fick's second laws (e.g., exponential, two-term, logarithmic, and Henderson and Pabis), and (c) first-order kinetics models (C. Ertekin & Firat, 2017; Krokida & Philippopoulos, 2005; Onwude, Hashim, Janius, Nawi, & Abdan, 2016)  [https://doi.org/10.1111/jfpe.13192]. It is also mentioned in literature that first-order models are more compatible to the experimental data of drying [https://doi.org/10.3390/pr10102082]. Because of the literature data and good results of first- order kinetic models in previous works of the authors [https://doi.org/10.1080/10498850.2021.1900969], first order kinetics were chosen.

4) Section Ultrasound and microwave assisted extraction (UMAE): authors must provide information (references, for example) to justify the choice of the operating conditions;

Regarding the operating condition of extraction time, the aim of the study is the application of “green” extractions and the use of environmentally friendly and energy-efficient conditions. Therefore, short extraction durations are chosen between 5 and 20 min to show that even during this short time the UMAE yield can be very high.

Regarding the ultrasonic power, the XO-SM50 Ultrasonic Microwave Reaction System reaches values up to 900 W. During our preliminary experiments, we performed extractions of 30 min and 900 W. These maximum values had a negative impact on phenolic content. Therefore lower power, namely 150, 450 and 750 W (16.7%, 50% and 83% of maximum), and lower duration (5, 10 and 20 min) were selected to be further studied, since they do not deteriorate the quality of the phenolic compounds. The decrease of the TPC when extreme powers and higher durations are applied is mentioned in other scientific papers [https://doi.org/10.1111/jfpp.14636, https://doi.org/10.1016/j.foodchem.2017.08.114,  https://doi.org/10.1016/j.lwt.2017.10.065]. The above-mentioned comments together with related literature to support our range of ultrasonic power utilized is added in the revised manuscript.

5) Section Calculation of the Extraction Yield (EY): provide the definition of TPC. In addition, the units in equation 4 (mg and g) must be in the same order of magnitude;

The definition of TPC is provided in Equation 4.  We agree with your comment and believe that the TPC and initial dry biomass should be in the same order of magnitude and this is what we have done in other papers, such as https://doi.org/10.1016/j.algal.2021.102374, . However since the mass of TPC is low in comparison with the initial dry biomass we chose to present mg/g so that the percentages of EY are more presentable and the mathematical modelling is carried out better. Please inform us if this is fine.

6) Section 3.1. Drying: provide a title for Table 3;

The caption of Table 3 is added.

7) Section 3.2.1. Extraction Yield (EY): in ” Table 3 demonstrates the estimated values of the mathematical model that was fitted to the experimental data.” the correct is Table 4. The numbering of the following tables should also be corrected;

The numbering of the Tables is corrected.

8) The results obtained should be compared with similar studies;

HPLC results and extraction yield results are compared with other research works studying the phytochemical profile of peach and apricot extracts after ultrasound assisted extractions.

9) Conclusion: indicate the best ultrasound power (W) for the operating condition.

The best performing ultrasound powers are presented in the results and in the conclusions.

Round 2

Reviewer 1 Report

The authors have revised the manuscript according to the reviewers' comment. No further comment from my end. 

Authors may consider English proofreading service.

Reviewer 3 Report

Authors made satisfactory improvements and responded appropriately to reviewers.

Minor revisions can be performed.